# Characterization of a Vesicular Stomatitis Virus-Vectored Recombinant Virus Bearing Spike Protein of SARS-CoV-2 Delta Variant

**DOI:** 10.3390/microorganisms11020431

**Published:** 2023-02-08

**Authors:** Wenwen He, Huan Cui, Shen Wang, Bo Liang, Cheng Zhang, Weiqi Wang, Qi Wang, Wujian Li, Yongkun Zhao, Tiecheng Wang, Zhuoran Liu, Bin Liu, Feihu Yan, Songtao Yang, Xianzhu Xia

**Affiliations:** 1Department of Laboratory Medicine, The Second Affiliated Hospital, Hengyang Medical School, University of South China, Hengyang 421001, China; 2Key Laboratory of Jilin Province for Zoonosis Prevention and Control, Changchun Veterinary Research Institute, Chinese Academy of Agricultural Sciences, Changchun 130122, China; 3College of Veterinary Medicine, Jilin University, Changchun 130062, China

**Keywords:** SARS-CoV-2, Delta variant, vesicular stomatitis virus, neutralization assay, recombinant virus

## Abstract

The frequent emergence of SARS-CoV-2 variants thwarts the prophylactic and therapeutic countermeasures confronting COVID-19. Among them, the Delta variant attracts widespread attention due to its high pathogenicity and fatality rate compared with other variants. However, with the emergence of new variants, studies on Delta variants have been gradually weakened and ignored. In this study, a replication-competent recombinant virus carrying the S protein of the SARS-CoV-2 Delta variant was established based on the vesicular stomatitis virus (VSV), which presented a safe alternative model for studying the Delta variant. The recombinant virus showed a replication advantage in Vero E6 cells, and the viral titers reach 10^7.3^ TCID_50_/mL at 36 h post-inoculation. In the VSV-vectored recombinant platform, the spike proteins of the Delta variant mediated higher fusion activity and syncytium formation than the wild-type strain. Notably, the recombinant virus was avirulent in BALB/c mice, Syrian hamsters, 3-day ICR suckling mice, and IFNAR/GR^−/−^ mice. It induced protective neutralizing antibodies in rodents, and protected the Syrian hamsters against the SARS-CoV-2 Delta variant infection. Meanwhile, the eGFP reporter of recombinant virus enabled the visual assay of neutralizing antibodies. Therefore, the recombinant virus could be a safe and convenient surrogate tool for authentic SARS-CoV-2. This efficient and reliable model has significant potential for research on viral-host interactions, epidemiological investigation of serum-neutralizing antibodies, and vaccine development.

## 1. Introduction

Severe acute respiratory syndrome coronavirus 2 (SARS-CoV-2) has rapidly led to a public health emergency of international concern since its discovery in December 2019 [1]. Emerging variants have garnered attention on a global scale. On 26 November 2021, the World Health Organization (WHO) categorized five SARS-CoV-2 variants as Variants of Concern (VOC), including Alpha (B.1.1.7), Beta (B.1.351), Gamma (P.1), Delta (B.1.617.2), and Omicron (B.1.1.529). The Delta variant was reported for the first time in India in October 2020 and has come under the spotlight due to its high pathogenicity and virulence [2]. It was reported that the Delta variant resulted in a higher risk of hospitalization, intensive care unit (ICU) admission, and mortality in COVID-19 patients than other VOCs [3,4]. In addition, data suggests that although the Omicron variant may have increased transmissibility, it caused less severe disease than the Delta variant [5]. Consequently, an in-depth study and full clarification of the characteristics of the most virulent SARS-CoV-2 variant is urgently needed.

Vesicular stomatitis virus (VSV) is a member of the rhabdovirus family with a single-stranded negative-stranded RNA nucleic acid encoding five structural proteins, including nucleoprotein (N), phosphoprotein (P), matrix protein (M), glycoprotein (G), and viral polymerase (L) [6,7]. Pseudotyped virus systems based on VSV have been established due to their strong operability, low biological risk, and convenient application [8]. Subsequently, VSV has successfully become a surrogate model for live virus, including Ebola virus (EBOV), Marburg virus (MARV), etc. [9] It was widely used in studies of virus entry mechanism and in the evaluation of neutralizing antibody activities and vaccine efficacy [10,11]. SARS-CoV-2 was classified as a high biological risk pathogen, and authentic viruses must be handled in a biosafety level 3 (BSL-3) laboratory which is not always accessible. Recently, a pseudotyped virus was used to simulate the biological characteristics of SARS-CoV-2 [11,12,13,14]. However, the pseudotyped virus cannot be amplified in vitro and needs to be rescued in real time, which brings experimental and economic pressure to the relevant research [8,10]. Previous studies have illustrated that a novel replication-competent VSV-based recombinant virus system was highly similar to authentic viruses that are suitable for the development of vaccine and neutralization assays [15,16,17]. The surface spike protein of SARS-CoV-2 was the main target for the induction of protective humoral and cellular immunity during SARS-CoV-2 infection [18]. Neutralizing antibodies were a major correlate of protective immunity and vaccine success [19]. This surrogate virus was safer, more consistent with the biological characteristics of live viruses, and more convenient for access and usage compared with available alternative approaches.

In this study, we reported a high titer of replication-competent recombinant vesicular stomatitis virus expressing the S proteins of the SARS-CoV-2 Delta variant (10^7.3^ TCID_50_/mL). It can simulate the authentic virus of cell tropism, cell damage, and viral replication kinetics. The recombinant virus could be handled in a biosafety level-2 (BSL-2) laboratory. Interestingly, we also noticed that rodents vaccinated with rVSVΔG-Sdel-eGFP produced a protective neutralizing antibody against SARS-CoV-2 Delta variant infection. Therefore, these results demonstrated that replication-competent rVSVΔG-Sdel-eGFP was a promising tool in future research on viral-host interactions, seroepidemiological studies, and vaccine development.

## 2. Materials and Methods

### 2.1. Animals and Ethics Statement

Three-day-old ICR mice, 6–8-week-old female BALB/c mice and Syrian hamsters were purchased from the Beijing Vital River Laboratory Animal Technology Company Limited. Six-to-eight-week-old male interferon-α/β and interferon-γ receptors double knockout mice (named IFNAR/GR^−/−^ mice) were purchased from The Jackson Laboratory. The animals were housed in accordance with the relevant guidelines and regulations. All of the animal experiments were carried out in BSL-2 facilities, following a protocol approved by the National Standards for Welfare and Ethics of Changchun Veterinary Research Institute, Chinese Academy of Agricultural Science (IACUC of AMMA-11-2020-20). All procedures involving authentic SARS-CoV-2 were conducted in a BSL-3 laboratory and approved by the animal experimental committee of the Laboratory Animal Center, Changchun Veterinary Research Institute, Chinese Academy of Agricultural Sciences. Animals were acclimatized for 3 days prior to infection, given food and water ad libitum and monitored daily. Environmental enrichment was also provided in the cages during the study.

### 2.2. Cells, Viruses and Antibodies

BSR/T7 cells, Huh7 cells, 293T cells, and Vero E6 cells were stored at the Changchun Veterinary Institute and cultured in Dulbecco’s Modified Eagle’s Medium (DMEM) (Gibco, Grand Island, NY, USA) supplemented with 10% fetal bovine serum (FBS; Sigma, St. Louis, MO, USA), and 1% penicillin/streptomycin (p/s) (Sigma, St. Louis, MO, USA) 37 °C under a 5% CO_2_ atmosphere. Replication-competent recombinant VSVs (rVSV-eGFP) (Indiana) and rVSV pseudotyped with Wuhan-Hu-1 isolate S protein (NC_045512.2) (rVSVΔG-WH01-eGFP) were generated and stored at the Changchun Veterinary Institute. The SARS-CoV-2 Delta variant (CSTR.16698.06. NPRC 6.CCPM-B-V-049-2105-8) was supplied by BSL-3 of Changchun Veterinary Institute. Rabbit anti-SARS-CoV-2 S polyclonal primary antibody was purchased from Sino Biological Inc (Beijing, China). CY3-conjugated goat anti-rabbit IgG (H+L) antibody and horseradish peroxidase (HRP)-conjugated goat anti-rabbit IgG (H+L) antibody were purchased from Beyotime Biotech Inc (Shanghai, China). Human serum samples, mouse polyclonal antibodies, and equine anti-SARS-CoV-2 serum/IgG were obtained from the Changchun Veterinary Institute.

### 2.3. Construction and Rescue of Recombinant Viruses

The VSV full-length plasmid was constructed in pcDNA3.1 vectors as described previously [17], named p3.1-VSV-eGFP. The glycoprotein protein gene of p3.1-VSV-eGFP was replaced by the SARS-CoV-2 Delta variant spike protein gene (OL336792.1), and the enhanced green fluorescence (eGFP) reporter gene was inserted between the N gene and the P gene to generate the recombinant full-length plasmid (designated as p3.1-VSVΔG-Sdel). The pcDNA3.1-VSV-N, pcDNA3.1-VSV-P, pcDNA3.1-VSV-L, and pcDNA3.1-VSV-G plasmids were generated as described previously [17], and stored at −80 °C. The p3.1-rVSVΔG-Sdel and four helper plasmids were cotransfected into BSR/T7 cells. The procedure was carried out in accordance with the calcium phosphate transfection kit instructions (Invitrogen, Waltham, MA, USA). The recovery of the viruses was determined by cytopathic effects and eGFP reporter expression. The first generation of the recombinant virus was passaged on Vero E6 cells (Figure 1).

### 2.4. Virus Titration

The Vero E6 cell lines were seeded in 96-well plates, and viral infection was carried out after the cells had reached a monolayer (2 × 10^4^ cells/well). A 10-fold serial dilution of recombinant viruses in 1.5 mL centrifuge tubes and the diluted viruses (100 µL/well) were added to 96-well cell culture plates with the cell supernatant discarded. Eight replicate wells were conducted at each dilution. The cell plates were moved to a 37 °C, 5% CO_2_ incubator for 1 h, and then added a 100 µL/well of DMEM containing 2% FBS. Finally, fluorescence was measured at 48 h post-infection (hpi) under an inverted fluorescence microscope (ZEISS), and the Reed-Muench method was used to calculate the 50% tissue culture infectious dose (TCID_50_).

### 2.5. Western Blotting

The rVSVΔG-Sdel-eGFP, rVSV-eGFP (negative control), and rVSVΔG-WH01-eGFP (positive control) were all mixed with protein loading buffers (CWBIO, Jiangsu, China), and the samples were boiled for 10 min. The denatured protein was separated using 10% SDS-PAGE gel electrophoresis (Beyotime), and the proteins were then wet-transferred to the nitrocellulose filter membrane (Beyotime) at 300 mA for 70 min. An amount of 50 g/L skim milk was incubated for 2 h at room temperature. Rabbit anti-SARS-CoV-2 S polyclonal primary antibody was diluted 1:3000 with 30 g/L skim milk and incubated at room temperature for 1.5 h. After five washes with PBST, HRP-conjugated goat anti-rabbit IgG was incubated with the membrane at room temperature for 1 h. Finally, the membrane was washed five times with PBST solution (phosphate-buffered solution containing 0.05% Tween-20) and examined with a Tanon 5200 chemiluminescence imaging system after the addition of electrochemiluminescence (ECL) immunoblotting substrate.

### 2.6. Immunofluorescence Analysis (IFA)

Vero E6 cells were seeded into 24-well plates, and infected with rVSVΔG-Sdel-eGFP or rVSV-eGFP when the cell density had reached 80–90%. The cells were then incubated for 1 h at 37 °C in an incubator supplied with 5% CO_2_. Subsequently, the cells were replenished with DMEM containing 2% FBS to 500 µL/well and incubated for 48 h. Virus-infected cells were fixed overnight at 4 °C with 4% paraformaldehyde and washed three times using phosphate-buffered saline (PBS). The rabbit anti-SARS-CoV-2 S polyclonal primary antibody (diluted 1:1000 with 1% BSA solution) was added to the washed plate (200 μL/well) and incubated at 37 °C for 1.5 h. The well was then washed three times with PBS solution and incubated with Cy3-labeled goat anti-rabbit IgG secondary antibody (diluted 1:1000 with 1% BSA solution) at 37 °C for 1 h. The nuclei were stained with 4,6-diamidino-2-phenylindole (DAPI) (diluted with 1% BSA solution at 1:1000) (Sigma, St. Louis, MO, USA) and incubated at 37 °C for 10 min. Finally, each well was washed three times with PBS solution. The fluorescence results were examined under a Zeiss inverted fluorescence microscope.

### 2.7. Transmission Electron Microscopy (TEM) Analysis

Recombinant viruses were negatively stained with uranyl acetate and then examined under an electron microscope. Briefly, the virus solution and β-Propiolactone were mixed and fixed with the copper-plated grids at room temperature. The grids were stained with 2% phosphotungstic acid solution (PTA) for 2 min at room temperature, after which excess liquid was removed. The grid was air-dried and observed under a transmission electron microscope.

### 2.8. Viral Replication Kinetics of Recombinant Viruses

Vero E6 cells were cultured on a 24-well plate overnight and infected with rVSVΔG-Sdel-eGFP or rVSV-eGFP at an MOI of 0.01, 0.1, and 0.5, respectively. The cell culture supernatant was harvested at 12, 24, 48, 60, 72, and 84 hpi, and the viral titer was then calculated using the Reed–Muench method. Finally, growth curves were outlined based on the viral titers at various MOI and time intervals.

### 2.9. Neutralizing Assay Using the Recombinant Viruses

The neutralizing antibody activity of COVID-19 vaccine-elicited human serum, mouse anti-SARS-CoV-2 polyclonal antibodies, and equine anti-SARS-CoV-2 serum/IgG using rVSVΔG-Sdel-eGFP and rVSVΔG-WH01-eGFP were tested; in previous experiments, the authentic virus (SARS-CoV-2 Delta variant) was used in the same conditions of the experiment. First, the samples were diluted in 96-well plates (initial dilution of 1:8 or 1:10, followed by two-fold gradient dilution), and the dilution was discarded up until the last well, and cell and viral controls were set up. The diluted sample was then mixed with 100 TCID_50_ of viruses (besides from cell controls) and incubated for 1 h at 37 °C in a 5% CO_2_ incubator. An amount of 2 × 10^4^ Vero E6 cells were added to each well, and then the fluorescence signal was observed after 48 h of incubation.

### 2.10. Safety Evaluation of rVSVΔG-Sdel-eGFP In Vivo

Five BALB/c mice and five Syrian hamsters were intraperitoneally (i.p.) injected with high doses of rVSVΔG-Sdel-eGFP (5 × 10^6^ TCID_50_ each BALB/c, 10^7^ TCID_50_ each Syrian hamster) at day 0 and monitored for survival, signs of illness and body weight changes for 10 consecutive days post-infection (dpi). Four mice deficient for type-I/II IFN receptor (IFNAR/GR^−/−^) were injected with 2 × 10^6^ TCID_50_ viruses per mouse via i.p. administration. Three-day-old ICR suckling mice were intracranially (i.c.) infected with 10^6^ TCID_50_ recombinant viruses, and the survival rate of the suckling mice was monitored for 10 consecutive days prior to sacrifice. Animals injected with rVSV-eGFP were used as a control group.

### 2.11. Animal Immunization and Challenge

For immunogenicity, at 6~8 weeks of age, two group of Syrian hamsters (nine per group) were intranasally immunized with either PBS (mock control) or rVSVΔG-Sdel-eGFP (2 × 10^6^ TCID_50_/animal) on day 0 and boosted with the same condition on day 15. Blood samples were collected at 22 days post vaccination to test vaccine-induced antibody response. At 0 days post infection (Day 25), all mice were transferred to the BSL-3 facility and were intraperitoneally challenged with the 10^5^ TCID_50_/animal SARS-CoV-2 Delta variant. Clinical signs and weight changes of these challenged mice were monitored daily. On 3 dpi, 5 dpi and 7 dpi, three mice in each group were euthanized, and equivalent portions of the nasal turbinate and lung tissues were collected for quantification of SARS-CoV-2 viral loads and RNA copies. Finally, the lungs of the Syrian hamsters were collected and analyzed by histology and immunohistochemistry.

### 2.12. RNA Extraction and RT-PCR Quantification

RNA extraction and RT-PCR quantification were performed according to a previous protocol [20]. Nasal turbinates and lung tissues were homogenized using an electric homogenizer for 300 s in 500 µL DMEM. The supernatant was collected and centrifuged at 12,000× *g* rpm for 10 min at 4 °C. The genome of the virus was extracted according to the manufacturer’s instructions (TIANGEN, Beijing, China), and viral RNA quantification was conducted by quantitative reverse transcription RNA (qRT-PCR) targeting the N gene of SARS-CoV-2. A qRT-PCR was performed with Premix Ex Taq (Takara, Beijing, China) with the following primers and probes: NF (5′-GGGGAACTTCTCCTGCTAGAAT-3′); NR (5′-CAGACATTTTGCTCTCAAGCTG-3′); and NP (5′-FAM-TTGCTGCTGCTTGACAGATT-TAMRA-3′).

### 2.13. Quantification of Viral Loads by TCID_50_

The TCID_50_ of nasal turbinates and lung homogenate supernatants were detected by the cytopathic effect (CPE), and the procedure was conducted as previously described [17]. The tissue homogenate supernatants were serially diluted in DMEM and added to Vero E6 cells (80–90% cell density) in a 96-well plate. The cell plates were moved to a 37 °C, 5% CO_2_ incubator for 1 h, and then added 100 µL/well of DMEM containing 2% FBS. The cytopathic effect was read 72 h later.

### 2.14. Histology and Immunohistochemistry (IHC)

Histology and IHC staining were performed following a previous protocol [18]. Briefly, the lung tissue fixed with 4% paraformaldehyde was sectioned at 5 μm, and then stained by hematoxylin and eosin (H&E) for histopathologic examination. For IHC sections, an anti-SARS-CoV-2 nucleocapsid antibody (Sino Biological Inc, Beijing, China) was used as the primary antibody to detect specific anti-SARS-CoV-2 immunoreactivity. The brown color represented immunoactivity in sections of the lungs.

### 2.15. Statistical Analysis

All data were analyzed using SPSS v25.0 software and the figures were plotted using GraphPad Prism v8.0 software. A Student’s *t*-test, two-way ANOVA analysis or Mann-Whitney test were used throughout this study for statistical analysis.

## 3. Results

### 3.1. Generation and Characterization of Replication-Competent Recombinant Virus Expressing SARS-CoV-2 Delta S Protein

The replication-competent recombinant virus expressing the SARS-CoV-2 Delta S protein was rescued. The p3.1-VSVΔG-Sdel plasmid was cloned into eukaryotic expression plasmid pcDNA3.1 to generate the full-length plasmid of SARS-CoV-2 Delta variant (named p3.1-VSVΔG-Sdel). The result showed a specific target band at roughly 3842bp by double enzyme digestion (Figure 2A). The p3.1-VSVΔG-Sdel and four helper plasmids pcDNA3.1-VSV-N, pcDNA3.1-VSV-P, pcDNA3.1-VSV-L, and pcDNA3.1-VSV-G were co-transfected into BSR/T7 cells, as described previously [17], and a replication-competent recombinant virus expressing SARS-CoV-2 Delta variant S protein was successfully rescued. Finally, viral genomes were extracted from the tenth passage virus to be verified by sequencing, and the result showed that no point mutations or insertion/deletion mutations were observed in the open reading frame of the S protein gene.

To determine the optimum cell line for rVSVΔG-Sdel-eGFP amplification and neutralization assays, we compared the viral titer in different mammalian cell lines. Among all the tested cell lines, Vero E6 cells (10^7.3^ TCID_50_/mL), 293T cells (10^5.667^ TCID_50_/mL), and Huh7 cells (10^6.492^ TCID_50_/mL) showed high amplification of the rVSVΔG-Sdel-eGFP, while BSR/T7 cells (10 TCID_50_/mL) were less so. Vero E6 cells were susceptible to the rVSVΔG-Sdel-eGFP and showed the highest level of virion production (Figure 2B), which was similar to previous studies [13,21]. As a result, Vero E6 cells were selected for the virus production and neutralization assays.

To explore whether the Delta strain differed in its ability to fuse at the plasma membrane, we compared the differences in fusion activity between rVSVΔG-Sdel-eGFP and rVSVΔG-WH01-eGFP in Vero E6 cells. Notably, the results indicated that the rVSVΔG-Sdel-eGFP-infected Vero E6 cells formed huge and oval cytoplasmic lesions with syncytia, while the cytopathic lesions of the rVSVΔG-WH01-eGFP displayed small and rounded lesions (Figure 2C). This phenomenon demonstrated that rVSVΔG-Sdel-eGFP infection exhibited an S protein-mediated entry and induced cell-cell fusion much more strongly than rVSVΔG-WH01-eGFP. In addition, several previous studies have demonstrated that the replication and fusion activity of the SARS-CoV-2 Delta variant is greatly enhanced in Vero E6 cells than in that of the wild-type SARS-CoV-2 or other variants, which is consistent with our findings [22,23,24,25]. One of the most important reasons causing this phenomenon was the critical mutations of the Delta variant, such as L452R, P681R, and D614G in the Delta S protein [26,27,28]. In conclusion, these results presented cell tropism profiles and cell-cell fusion of the rVSVΔG-Sdel-eGFP that was similar to what was caused by the SARS-CoV-2 Delta variant.

### 3.2. Identification of the Recombinant Virus

To detect whether recombinant viruses rVSVΔG-Sdel-eGFP can express the S protein of the SARS-CoV-2 Delta variant, immunoblotting was used to determine the status of SARS-CoV-2 S protein expression in the supernatant of Vero E6 cells infected with fifth- and tenth-generation recombinant viruses. The results showed a clear target band at approximately 190 KDa that indicated that the S protein appeared in both generation virions and virus-infected cell lysates (Figure 3A). Furthermore, the expression of the SARS-CoV-2 S protein was studied by immunofluorescence 48hpi of Vero E6 cells with either rVSVΔG-Sdel-eGFP or not (Figure 3B). The antibody specific for SARS-CoV-2 S protein reacted with cells infected with rVSVΔG-Sdel-eGFP, whereas the negative control did not bind. These results confirmed the efficient expression of SARS-CoV-2 S protein on the surface of the VSV-based recombinant virus.

To analyze whether exogenous envelope protein genes affect the structure of recombinant viruses, the parental VSV and the rVSVΔG-Sdel-eGFP morphological changes in cell culture supernatant were observed under transmission electron microscopy (Figure 3C). The rVSVΔG-Sdel-eGFP showed a typical bullet shape and was approximately 190 nm, consistent with previous reports [29]. Compared to the parental VSVs, the spikes on the surface of recombinant virus became thickened and coronal in distribution. These results suggested that the S protein has efficiently packaged on the surface of rVSVΔG-Sdel-eGFP.

### 3.3. Growth Kinetics of Recombinant Viruses

To investigate the optimal production conditions and growth kinetics of the recombinant viruses, the growth curve of rVSVΔG-Sdel-eGFP was determined in Vero E6 cells at an MOI of 0.5, 0.1, and 0.01, respectively (Figure 3D). The one-step growth curve of rVSVΔG-Sdel-eGFP was similar, with a parental VSV at an MOI of 0.5, 0.1, and 0.01, respectively. At an MOI of 0.01, the titers of rVSVΔG-Sdel-eGFP and parental VSV at 36 hpi were the highest and could reach 10^7.3^ TCID_50_/mL and 10^9.7^ TCID_50_/mL, respectively. However, compared with parental VSV, the titer of rVSVΔG-Sdel-eGFP reduced 1000-fold, probably because of the insertion of a large foreign gene. Although the insertion of the exogenous envelope protein gene renders the packaging of the recombinant virus less efficient, our results suggested that the highest viral titer of the recombinant virus could reach 10^7.3^ TCID_50_/mL, which was higher than other VSV pseudoviruses of SARS-CoV-2 [30,31]. Therefore, the optimal culture condition for rVSVΔG-Sdel-eGFP was cultured at 37 °C for 36 h in Vero E6 cells with an MOI of 0.01.

### 3.4. Safety Evaluation of rVSVΔG-Sdel-eGFP In Vivo

To investigate the safety of rVSVΔG-Sdel-eGFP, we performed a safety evaluation in BALB/c adult mice, Syrian hamsters, ICR suckling mice, and IFNAR/GR^−/−^ mice. The survival rate and body weight changes of all virus infected animals were monitored daily. The parental VSV (rVSV-eGFP) served as control (Figure 4A). All of the BALB/c mice infected rVSVΔG-Sdel-eGFP and rVSV-eGFP showed a 100% survival rate until the end of the experiment, and none of them showed body weight loss (Figure 4B,C). In contrast, Syrian hamsters infected with rVSV-eGFP showed sustained weight loss, signs of depression, unruly hair, and hind limb paralysis at 3 dpi, and even showed death at 6 dpi, consistent with previous studies [9,32] (Figure 4D,E). However, all Syrian hamsters injected with the recombinant virus survived without adverse clinical symptoms. The mechanism was unclear and needed to be further determined in the future study. Three-day-old ICR suckling mice injected with recombinant virus (10^6^ TCID_50_ per mice) survived the observation period, but all of the suckling mice challenged with rVSV-eGFP succumbed from 2 dpi to 4 dpi (Figure 4F). Interestingly, IFNAR/GR^−/−^ mice inoculated with 2 × 10^6^ TCID_50_/animal rVSVΔG-Sdel-eGFP survived, while the control group was dead at 2 dpi (Figure 4G). Based on these results, the recombinant viruses rVSVΔG-Sdel-eGFP preliminarily showed excellent safety in BALB/c adult mice, Syrian hamsters, ICR suckling mice, and IFNAR/GR^−/−^ mice.

### 3.5. Establishment of a Surrogate Virus Neutralization Assay

To determine whether the rVSVΔG-Sdel-eGFP expressing the SARS-CoV-2 Delta S protein could serve as a surrogate and valid model for assessing neutralizing antibody activities, a recombinant virus neutralization assay was established using vaccine-elicited human serum, mouse anti-SARS-CoV-2 polyclonal antibodies and equine anti-SARS-CoV-2 serum/IgG. These sera and antibodies were tested for their ability to neutralize the rVSVΔG-Sdel-eGFP, in comparison with the rVSVΔG-WH01-eGFP and authentic virus (SARS-CoV-2 Delta variant). We found that there was no significant difference between the detection results of neutralizing antibodies of recombinant viruses and those of authentic SARS-CoV-2 (Mann-Whitney test, *p* > 0.05) [33], but our data showed that the neutralization activity of sera or IgG against Delta variants was significantly reduced compared to the rVSVΔG-WH01-eGFP (Mann-Whitney test, *p* = 0.002, Figure 5A–C). Previous studies showed that the neutralization activity of monoclonal antibodies, convalescent plasma, and vaccine against the Delta strain were reduced compared with that of the wild-type SARS-CoV-2 [34,35], consistent with our data. Furthermore, we determined the extent to which the rVSVΔG-Sdel-eGFP and SARS-CoV-2 Delta variant neutralization tests correlated with each other. There was a high correlation between the two assays (*r* = 0.95, *p* < 0.0001) (Figure 5D). In addition, the reliability of using recombinant VSV instead of the authentic virus to detect SARS-CoV-2 specific neutralizing antibodies has been shown in other studies [16,21,33]. These results indicate that the rVSVΔG-Sdel-eGFP, which expresses the SARS-CoV-2 Delta spike protein, can substitute authentic SARS-CoV-2 for neutralizing antibody detection in various serum or antibody samples.

### 3.6. The rVSV Protected Syrian Hamsters against the SARS-CoV-2 Delta Variant Challenge

To explore whether rVSVΔG-Sdel-eGFP has the potential to be a potent live vector vaccine against the SARS-CoV-2 Delta variant, we tested neutralizing antibody (NAb) titers in Syrian hamsters. For VSV-vectored replication-competent COVID-19 vaccine candidates, it was reported that intranasal inoculation would be a more promising delivery route [17]. Therefore, Syrian hamsters were immunized with the rVSVΔG-Sdel-eGFP by intranasal administration on day 0 and day 15. Serum samples were collected after two-doses of rVSVΔG-Sdel-eGFP (on day 22), and SARS-CoV-2 neutralizing antibodies were detected using a surrogate virus neutralization assay, as has been described (Figure 6A). The results indicated that the geometric mean titers (GMTs) of serum NAbs in rVSVΔG-Sdel-eGFP-vaccinated mice totaled 796 (reciprocal serum titer), and were significantly increased compared to Syrian hamsters vaccinated with control PBS (*p* < 0.001) (Figure 6B).

Subsequently, the vaccine protective efficacy was tested in the Syrian hamster model challenged with the SARS-CoV-2 Delta variant. ten days after the booster (Day 25), all Syrian hamsters were i.p. challenged with 10^5^ TCID_50_ SARS-CoV-2 Delta strain. Clinical symptoms, survival rate, and body weight were monitored for a week post-challenge, and the results showed that both groups of mice survived to the end of the experiment (Figure 6C). However, the survival rate of PBS-vaccinated Syrian hamsters was 100%, which is consistent with the rVSVΔG-Sdel-eGFP-vaccinated group, and the body weight was significantly decreased in the PBS-vaccinated Syrian hamsters (all *p* < 0.05) (Figure 6D).

In addition, nasal turbinate and lung tissues were investigated for viral load and viral RNA at 3 dpi, 5 dpi and 7 dpi, respectively. The findings demonstrated that the viral RNA and viral titer in both the rVSVΔG-Sdel-eGFP-vaccinated group and the control group of mice gradually decreased over the time of the challenge, yet there was a significant difference between the two groups (Figure 6E–H). The viral RNA copies and viral loads in the lungs of PBS-vaccinated groups at 3 dpi, with peak values of ~10^6.73^ copies/mL and 10^3.58^ TCID_50_/mL respectively, were significantly higher than the rVSVΔG-Sdel-eGFP-vaccinated group (*p* < 0.01, Figure 6E,F). Although the viral RNA copies and viral titers of lung tissue between two groups had no statistical difference at 7 dpi, the rVSV-vaccinated group was 3.2 logs and 25 TCID_50_ lower than the control, respectively (Figure 6E). Likewise, the turbinate tissues of rVSV-vaccinated animals presented a significant reduction of the viral load and viral titer in lungs and nasal turbinates compared to PBS control (all *p* < 0.05, Figure 6G,H).

Finally, the lung tissues at 7 dpi from immunized and challenged Syrian hamsters were collected and stained with hematoxylin, eosin (H&E), and IHC (Figure 7). Consistent with the viral load data, lung tissue obtained from PBS-vaccinated mice at 7 dpi showed the consolidation of pulmonary parenchyma and inflammatory cell infiltration in H&E sections. However, rVSV-vaccinated Syrian hamsters were all protected from these pathologies, and demonstrated normal alveolar bronchial function (Figure 7A,B). In addition, immunoactivities in lung tissue in PBS-vaccinated mice were observed, but not for rVSV-vaccinated mice (Figure 7C,D). These data suggested that the rVSVΔG-Sdel-eGFP has potential as an immunogen, as it stimulated the rodents to generate an excellent immune response and induced robust protection from SARS-CoV-2 Delta strain infection.

## 4. Discussion

Accompanied with the evolution process, SARS-CoV-2 has undergone constant mutations, which may ultimately enhance “viral fitness” and selective adaptation. The Delta variant of SARS-CoV-2 has spread across at least 100 countries and had a high predominance rate in most of them. The Delta variant had higher transmissibility and patients infected with the Delta variant had greater viral loads, risk of hospitalization and ICU admission [2,3]. Although increased transmissibility and were dominant in current days, Omicron variants decreased mortality than other variants [36,37]. Significant uncertainty caused by the emerging variants existed in healthy humans due to the limited amount of data available. Thus, actions must be taken to combat the virulent SARS-CoV-2 variants, especially the Delta variant. To clarify the characterization of Delta variants, a safe surrogate model which could be handled in a BSL-2 laboratory was needed.

Recombinant virus rVSVΔG-Sdel-eGFP shares several common characteristics of the authentic virus. The virus entry, receptor binding, and membranous fusion of recombinant virus were determined by a spike protein rather than a VSV glycoprotein [8,38]. Accordingly, recombinant viruses could simulate the natural infection process of the authentic virus, consequently rendering it an alternative research tool for the researching of SARS-CoV-2. Our data suggest that the recombinant virus replicates stably in Vero E6 cells, as the viral titer peaked 36 h post incubation at an MOI of 0.01, reaching an upmost titer of 10^7.3^ TCID_50_/mL. Compared with recombinant VSV vectored virus harboring S protein from other SARS-CoV-2 strains, the rVSVΔG-Sdel-eGFP shows faster growth kinetics and a higher titer [21,39,40]. Furthermore, rVSVΔG-Sdel-eGFP induced increased cell-cell fusion and syncytia compared with the rVSVΔG-WH01-eGFP, which is consistent with a previous report [24]. The above phenomena may help to explain the increased virulence of Delta variants compared with other variants. In addition, the replication-competent recombinant virus carrying eGFP reporters is a powerful tool for the evaluation of NAbs since a high correlation between VSV-vectored recombinant virus and authentic virus has been confirmed [13,33,41]. The replication-competent recombinant virus would be a convenient and feasible choice of a high-cost effect compared with traditional pseudoviral systems. Based on the neutralizing assay we established, a reduced neutralizing ability against the Delta variant compared with the wild-type strain was observed, which is consistent with previous findings [34,35]. The above biological properties of recombinant virus support the future application of rVSVΔG-Sdel-eGFP as a safe alternative model of authentic viruses. Recently, the S proteins with a truncation in the cytoplasmic domain of coronaviruses were demonstrated to increase efficiency regarding incorporation into virions of VSV [28,31]. However, to ensure the authenticity of the spike protein phenotypes being investigated, the full-length of the SARS-CoV-2 Delta S protein was incorporated into VSV vectors. Of interest, a high titer of rVSVΔG-Sdel-eGFP up to 10^7.3^ TCID_50_/mL was registered. It was even higher than the titer of recombinant VSV viruses that bared the partial deletion of the spike glycoprotein cytoplasmic domain [30,31], which benefitted the production of the VSV-vectored COVID-19 vaccine. 

Preclinical evaluation in COVID-19 animal models would provide preliminary data on the safety and efficacy of vaccines, antibodies and chemical drugs. Currently, animal models including mice, golden hamsters, ferrets and nonhuman primates (NHPs) are involved in the study of COVID-19. Mice are not susceptible to SARS-CoV-2. Consequently, strategies such as hACE2 knock-in (transgene), hACE2 transduced, or mice-adapted SARS-CoV-2 were conducted to improve susceptibility [42]. A high viral load in the respiratory tract and a fatality rate of up to 100% were observed in these mouse models [43,44,45,46,47,48,49,50,51]. Nevertheless, the relationship between these artificially introduced hACE2 or adaption and SARS-CoV-2 infection in humans is largely unknown. Ferrets are naturally susceptible to SARS-CoV-2 and they recapitulate some typical disease features of COVID-19 observed in humans. Naturally infected ferrets rapidly transmitted SARS-CoV-2 to the entire population through direct or indirect contact [52,53]. Ferrets represent an ideal animal model for viral shedding, transmission, and post-infection transcriptional response. However, due to the relatively low viral titers in the lungs as well as the inconvenience in operation, the application of ferret models was reported less [54]. Non-human primates (NHPs) showed similar biological characteristics to humans, which were seen as gold standard models for many emerging infectious diseases. Rhesus macaques and cynomolgus macaques were permissive to SARS-CoV-2 infection and displayed COVID-19-like disease, whereas NHPs are not always available [55,56,57,58,59,60]. In contrast, golden hamsters are naturally susceptible to SARS-CoV-2, which represents a mild/moderate model of SARS-CoV-2 infection [61]. Weight loss, rapid breathing, and viral replication in the respiratory tract were observed in SARS-CoV-2 inoculated hamsters. The transmission of SARS-CoV-2 between hamsters was accomplished through direct contact or aerosols [62,63]. Therefore, golden hamsters are economic, easy to use, and an accessible small animal model for COVID-19. In this study, proof of concept evaluation of rVSVΔG-Sdel-eGFP as an immunogen was conducted in golden hamsters. According to preliminary exploration of the immunogenicity of the recombinant virus, a significantly increased immune response against the Delta variant was observed in rodents after two doses of the recombinant virus vaccination. In comparison to a mock control group, a reduction in virus load and viral RNA copies and a reduction in virus detection in the lungs and nasal turbinates of rVSV-immunized Syrian hamsters, indicating rVSVΔG-Sdel-eGFP, could be used as a potent immunogen. Interestingly, compared with rVSVΔG-Sdel-eGFP-vaccinated hamsters, despite a high level of weight loss, a viral load and RNA copies could be detected in PBS-vaccinated hamsters after being challenged with the SARS-CoV-2 Delta variant; the titer of neutralizing antibody is increased in PBS-vaccinated hamsters. Therefore, changes in body weight and viral load may be better indicators of SARS-CoV-2 challenge.

Our study has several limitations. Although we have proposed some issues that may help to explain the increased pathogenicity of the SARS-CoV-2 Delta variant, further investigations are needed to uncover the inherent pathogenesis. In addition, we provided a preliminary evaluation of rVSVΔG-Sdel-eGFP as an immunogen, which did stimulate a robust immune response against the Delta variant in rodents. However, whether it can be used as an effective vaccine to stimulate the production of high titers of neutralizing antibodies in small animals against the currently prevalent Omicron variant strain requires further investigation. Finding COVID-19 vaccine targets that stimulate the production of more broadly neutralizing antibodies in the organism and better immunization strategies may be important to effectively control the SARS-CoV-2 pandemic. 

Taken together, a replication-competent rVSV carrying the SARS-CoV-2 Delta spike protein was developed in this study, which was presented as an effective tool for studying the SARS-CoV-2 Delta variant in a BSL-2 facility. The biological characteristics of the recombinant viruses such as cell tropism and replication growth curve were consistent with authentic viruses, and the high titer of up to 10^7.3^ TCID_50_/mL was observed in susceptible cells. We also demonstrated that the recombinant virus elicited strong immune response in rodents as an excellent immunogen; these responses protected the Syrian hamsters from the SARS-CoV-2 Delta variant challenge. Therefore, a replication-competent rVSV would be of significance in the study of virus-host interactions, neutralizing antibody detection, COVID-19 vaccine production, and even providing references for the future study of other emerging SARS-CoV-2 variants.

## Figures and Tables

**Figure 1 microorganisms-11-00431-f001:**
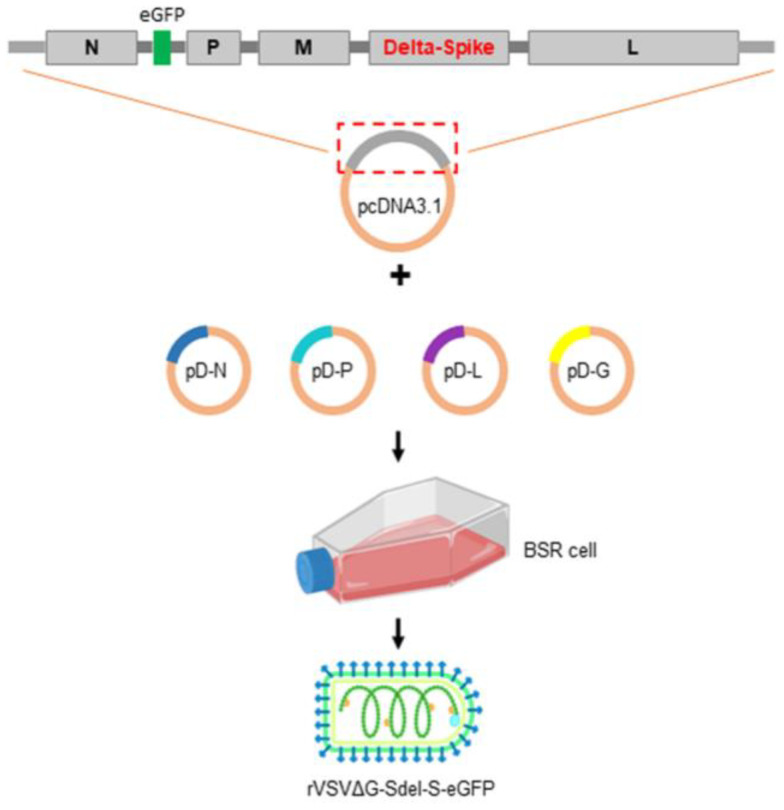
Scheme diagram of the generation of recombinant vesicular stomatitis virus rVSVΔG-Sdel-eGFP. The p3.1-VSV-eGFP plasmid in which its native glycoprotein gene was replaced with the SARS-CoV-2 Delta variant S protein gene to generate the recombinant full-length plasmid (named p3.1-VSVΔG-Sdel). The recombinant full-length plasmid and four helper plasmids were cotransfected into BSR/T7 cells and obtained a recombinant virus bearing the Delta strain S protein.

**Figure 2 microorganisms-11-00431-f002:**
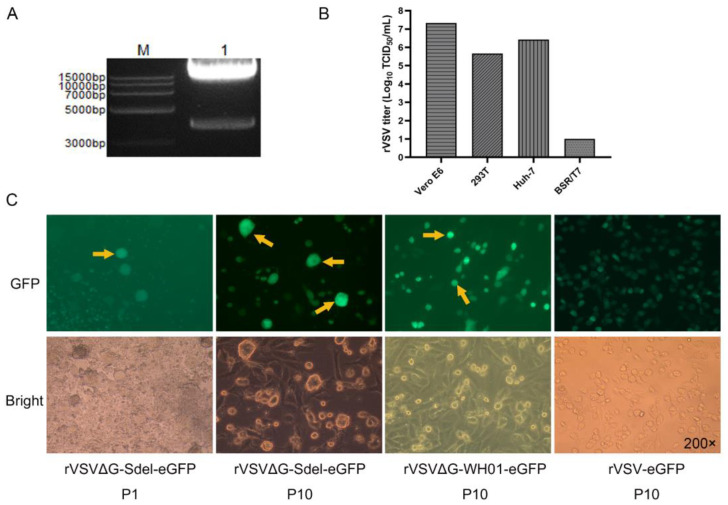
Generation and characterization of rVSVΔG-Sdel-eGFP. (**A**) Enzymatic digestion identification of the pcDNA3.1-VSVΔG-Sdel plasmid. (**B**) Infectivity of rVSVΔG-Sdel-eGFP on four types of cell lines. Infectivity was quantified by testing TCID_50_ in the above cell lines. (**C**) Diagram of cytopathic differences between rVSVΔG-Sdel-eGFP, rVSVΔG-WH01-eGFP and rVSV-eGFP in BSR/T7 cells (P1) and Vero E6 cells (P10). P represents passage.

**Figure 3 microorganisms-11-00431-f003:**
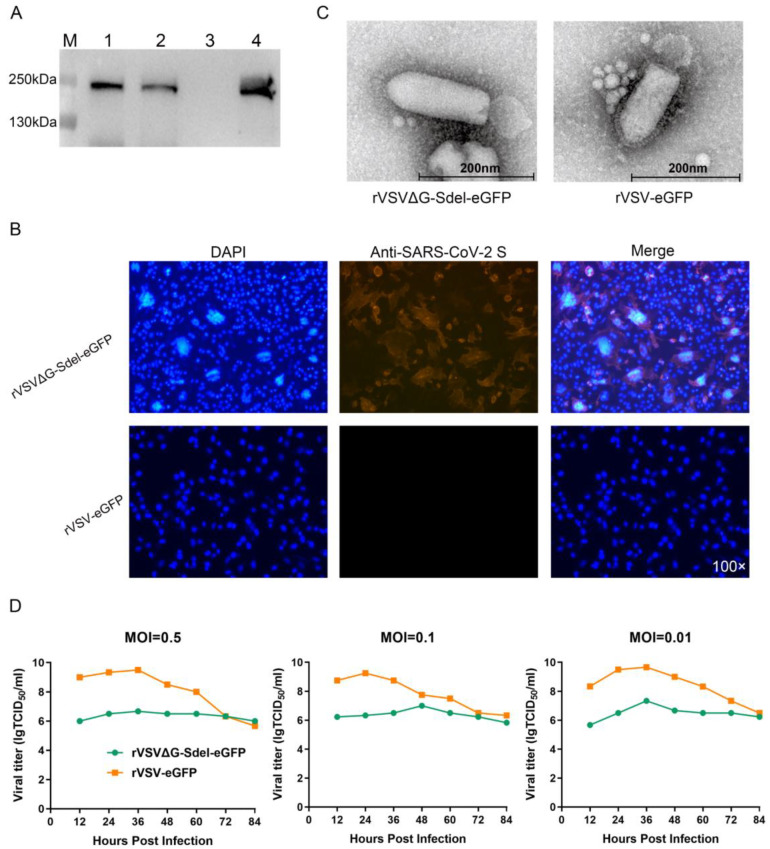
Properties of recombinant viruses rVSVΔG-Sdel-eGFP. (**A**) Western blot of spike protein in rVSVΔG-Sdel-eGFP from the indicated recombinant virus produced in Vero E6 cells 48 hpi. 1, 2 represented passage 5 and passage 10 of rVSV; 3, 4 represented the cell control and the positive virus control (rVSVΔG-WH01-eGFP). (**B**) IFA analysis of the S protein expression of the SARS-CoV-2 Delta variant. (**C**) The morphology of the recombinant virus under transmission electron microscopy. The scale bar was equivalent to 200 nm. (**D**) One-step growth curves of recombinant virus. rVSVΔG-Sdel-eGFP and rVSV-eGFP were inoculated into Vero E6 cells at an MOI of 0.5, 0.1, and 0.01, respectively. Viruses were collected at 12 h intervals to measure titers.

**Figure 4 microorganisms-11-00431-f004:**
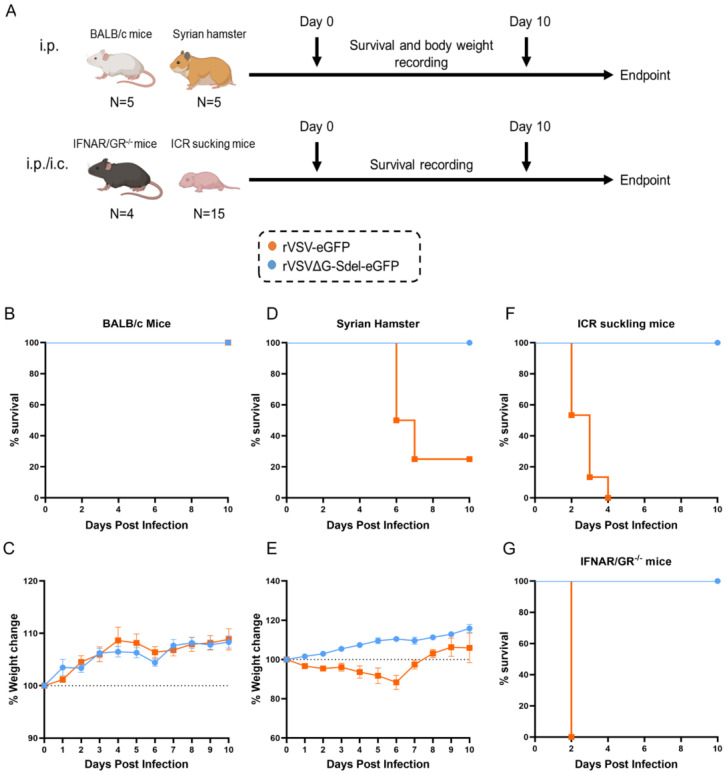
Body weight changes and survival curve of animals post rVSVΔG-Sdel-eGFP infection. (**A**) Safety evaluation procedure of animal. Animals were intraperitoneally or intracranially administrated with rVSVΔG-Sdel-eGFP on Day0. Survival rate and body weight changes of animals were monitored for 10 consecutive days post-infection. (**B**,**C**) The survival rate and weight change of BALB/c mice (*n* = 5). (**D**,**E**) The survival rate and weight change of Syrian hamsters (*n* = 5). (**F**) The survival rate of ICR suckling mice (*n* = 15). (**G**) The survival rate of IFNAR/GR^−/−^ mice (*n* = 4). rVSV-eGFP infection served as the control group. Data represent the mean ± standard error of the mean (SEM). The dotted line indicates the percentage of initial weight.

**Figure 5 microorganisms-11-00431-f005:**
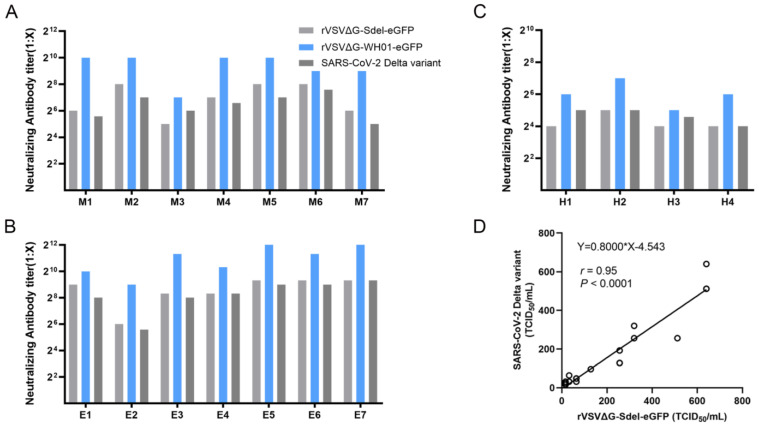
The application of recombinant viruses in serological assays. (**A**–**C**) Neutralizing antibody activities in mice (**A**), equine (**B**) and human (**C**) samples were determined using a surrogate virus neutralization assay. The rVSVΔG-WH01-eGFP and authentic virus (SARS-CoV-2 Delta variant) were used as a control virus. Each sample was repeated twice in parallel, and the mean is shown in the graph. M, mice; E, equine; H, human. (**D**) Correlation analysis of neutralization TCID_50_ values between rVSVΔG-Sdel-eGFP and the SARS-CoV-2 Delta variant.

**Figure 6 microorganisms-11-00431-f006:**
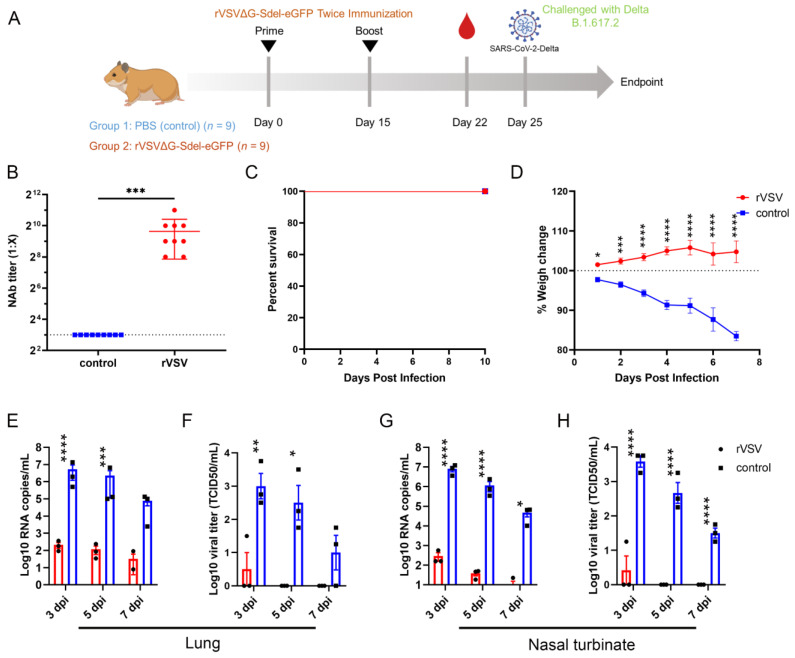
Immunogenicity evaluation of recombinant viruses in Syrian hamsters. (**A**) Immunization and challenge strategy. Syrian hamsters (nine animals per group) were intranasally immunized with recombinant viruses (2 × 10^6^ TCID_50_/animal) or PBS (control). Animals were boosted at a 15-day interval. Blood samples were collected at 22 days post vaccination. On day 25, Syrian hamsters challenged 10^5^ TCID_50_/animal SARS-CoV-2 Delta variants. (**B**) The titer of neutralizing antibodies against rVSVΔG-Sdel-eGFP in Syrian hamsters. (**C**) Survival rate of Syrian hamster. (**D**) Weight changes of Syrian hamsters. (**E**–**H**) Viral RNA copies (**E**,**G**) and viral titer (**F**,**H**) of lungs and turbinate samples. A statistical analysis was determined using a Student’s *t*-test or two-way ANOVA analysis. * *p* < 0.05, ** *p* < 0.01, *** *p* < 0.001, **** *p* < 0.0001.

**Figure 7 microorganisms-11-00431-f007:**
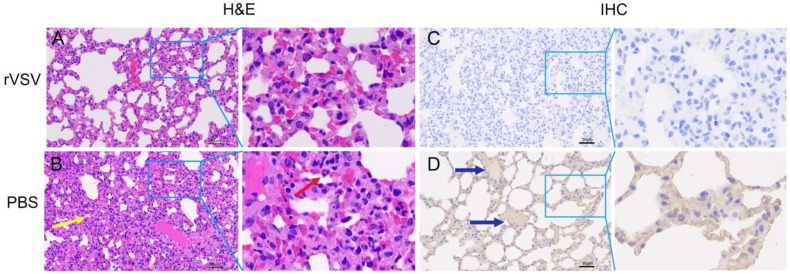
Lung pathology was reduced in rVSVΔG-Sdel-eGFP-vaccinated Syrian hamsters following challenge with the SARS-CoV-2 Delta variant. (**A**–**D**) The histopathology and immunohistochemistry (IHC) of Syrian hamster lungs are shown. (**A**) H&E stained sections of lung tissues obtained from rVSV-vaccinated mice shows no pathology after challenge. (**B**) The lungs of PBS-immunized mice showed marked pulmonary lesions, including severe parenchyma inflammation (the red arrow) and consolidation of pulmonary parenchyma (the yellow arrow) in H&E staining sections. (**C**,**D**) Numerous immunoreactivities (the blue arrow) are observed in the lung of the mock group, but no immunoreactivity was observed in the rVSV-vaccinated group. The scale bar = 50 μm.

## Data Availability

Not applicable.

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
