# Peer review of "Characterization of a Vesicular Stomatitis Virus-Vectored Recombinant Virus Bearing Spike Protein of SARS-CoV-2 Delta Variant"

_microorganisms, 2023, doi:10.3390/microorganisms11020431_

Round 1
Reviewer 1 Report
vectored Recombinant Virus Bearing Spike Protein of SARS-CoV-2 Delta Variant” submitted by He et al reports the development and characterization of a recombinant virus that deploys the spike protein of SARS-CoV-2 Delta variant. Authors describe the characteristics of their S-delta recombinant virus regarding in vitro and in vivo properties. The paper purposes an interesting and useful potential tool for viral-host studies, neutralization surrogate assay, and vaccine development, however, several major concerns arose, and these are listed as follows:
Major comments
-Line 131, describe what is the positive control, the source, or their identity.
-Line 178, see the comment for lines 313-328.
-Line 193, what is the Delta variant that was used? The source? This is related too with lines 307-308, the western blot assay contains this positive control, therefore it is necessary to provide this information.
-Line 268, a control with cells infected with rVSV-eGFP (Indiana) is lacking.
-Line 308, given that the paper's aim is to obtain a tool that in some experiments it is proposed as a surrogate virus of SARS-CoV-2 Delta variant, this experiment should be performed with the authentic virus along with the recombinant viruses.
-Lines 313-328, I disagree that this assay is named “Evaluation of the pathogenicity…” because to test this, it is necessary to use animal models in which the “authentic virus” causes the illness to demonstrate the differences or similarity with the surrogate one. Even, at the end of the paragraph the authors closed by saying that “…recombinant viruses….showed excellent safety in BALB/c adult mice, Syrian…” mixing the concepts pathogenicity, and safety. My guess is that this assay is a safety test of the recombinant viruses, given that no extra information is given regarding the SARS-CoV-2 Delta variant used, therefore I assume it is wild-type virus, not a mouse-adapted strain that could cause illness or lethality in BALB/c mice (e.g. Ref. PMID 35023830 and 35900097).
How can be explained that the control groups treated with recombinant virus expressing GFP only are lethal? If VSV causes this effect normally in these models, this should be stated if not, this must be addressed because this would suggest that the insertion of the S sequence abrogated this lethality.
-Lines 338-352 and 358-359, this assay holds the same weakness as the immunofluorescence (Figure 2B), the authors claim that this delta recombinant virus could be used as a surrogate virus of the authentic virus variant, even though they stated that “there was no significant difference between the detection results of neutralizing antibodies of two viruses and those of authentic SARS-CoV-2 and Delta variant” (line 343-344), they did not mention and not show the results with the authentic virus, they used a recombinant virus expressing Wuhan variant as a control virus. However, to warrant the neutralization assays are equivalent, they must run side-by-side their recombinant viruses with the authentic virus. Furthermore, the authors supported their claim that this recombinant virus is a surrogate for this assay because they detected lesser titers of antibodies in the samples tested (from humans, mice, and horses) for delta recombinant virus in comparison with Wuhan recombinant virus and this finding is accordingly with other reports, we do not exactly how these results could be using the authentic virus.
-Line 379, the lack of lethality in the challenge assay in Syrian hamsters, which is a very common infection model described as SARS-CoV-2 infection, weakness the findings in this part of the paper, because one of the possibilities was to use this recombinant virus as a potential vaccine and full potentiality could not be tested, this should be further discussed.
Minor comments
-Line 47, develop the acronym ICU.
-Lines 57-58, develop the acronyms EBOV and MARV.
-Line 124, develop the acronym EP.
-Line 134, develop the acronym NC.
-Line 135, what does “closed” mean? I think “incubated” is a better option.
-Line 135, describe the way that the polyclonal antibody was obtained.
-Line 137, given that the acronym HRP was developed at line 106, in this line “horseradish peroxidase ( )” needs to be deleted.
-Line 140, the revealing substrate is missing (I assume is a reagent with luminol).
-Line 145, a capital letter is missing after “48h.”
-Line 165-166, I consider “calculated” a more appropriate word instead of “detected”.
-Line 168, given that the neutralizing assays are performed to test the functional activity of antibodies, it is more suitable to say, “Neutralizing assay using the recombinant viruses”.
-Line 193, I consider that “variant” is the proper word than “strain”.
Author Response
Response to Reviewer 1 Comments
Thank you for spending time in reviewing our manuscript and providing us with a list of constructive comments. Those comments are all valuable and very helpful for revising and improving our paper, as well as the important guiding significance to us. We have studied comments carefully and have made correction which we hope meet with approval.
Here are the responses to reviewer 1:
[Major comments]
Point 1: Line 131, describe what is the positive control, the source, or their identity.
Response 1: Thanks for your comments. Positive control has been defined in line135-136. The positive control is recombinant VSV expressing the S protein of Wuhan-Hu-1 strain and the negative control is Vero E6 cell. The rVSVΔG-WH01-eGFP was generated and stored at the Changchun Veterinary Institute. (Lines 102-103, 135-136)
Point 2: Line 178, see the comment for lines 313-328.
Response 2: “Evaluation of the pathogenicity of ….” has been changed to “Safety evaluation of …” (Line 191)
Point 3: Line 193, what is the Delta variant that was used? The source? This is related too with lines 307-308, the western blot assay contains this positive control, therefore it is necessary to provide this information.
Response 3: We apologize for not providing a detailed information of the Delta variant. The information on the Delta variant has been provided in the part of “Materials and Methods”. The SARS-CoV-2 Delta variant (CSTR.16698.06. NPRC 6.CCPM-B-V-049-2105-8) was supplied by BSL-3 of Changchun Veterinary Institute. (Lines 104-105)
Point 4: Line 268, a control with cells infected with rVSV-eGFP (Indiana) is lacking.
Response 4: Pictures of cells infected with rVSV-eGFP (Indiana) have been provided in Figure 2C. (Line 271)
Point 5: Line 308, given that the paper's aim is to obtain a tool that in some experiments it is proposed as a surrogate virus of SARS-CoV-2 Delta variant, this experiment should be performed with the authentic virus along with the recombinant viruses.
Response 5: We regret that the parallel comparison was not conducted due to the limitation of our study design. According to your suggestion, we have already supplemented the relevant experiments and the data have been presented in lines 368-371 and Figure 4E. Simultaneously, there are abundant studies to support our data. (PMID:32738193, PMID:32990161, PMID:32735849)
Point 6: Lines 313-328, I disagree that this assay is named “Evaluation of the pathogenicity…” because to test this, it is necessary to use animal models in which the “authentic virus” causes the illness to demonstrate the differences or similarity with the surrogate one. Even, at the end of the paragraph the authors closed by saying that “…recombinant viruses….showed excellent safety in BALB/c adult mice, Syrian…” mixing the concepts pathogenicity, and safety. My guess is that this assay is a safety test of the recombinant viruses, given that no extra information is given regarding the SARS-CoV-2 Delta variant used, therefore I assume it is wild-type virus, not a mouse-adapted strain that could cause illness or lethality in BALB/c mice (e.g. Ref. PMID 35023830 and 35900097).
Response 6: Thanks for your advice. We are very sorry that we have mixed up two concepts. It is indeed a safety test of the recombinant viruses in this study. The title of this section has been changed to “Safety evaluation of rVSVΔG-Sdel-eGFP in vivo”. (Line 326)
Point 7: How can be explained that the control groups treated with recombinant virus expressing GFP only are lethal? If VSV causes this effect normally in these models, this should be stated if not, this must be addressed because this would suggest that the insertion of the S sequence abrogated this lethality.
Response 7: Firstly, the parental VSV (Indiana strain) was found to be pathogenic to hamsters in previous studies (PMID: 32842671,PMID: 2414943,PMID: 6265369), and our results are consistent with their findings. These studies found that hamsters were exquisitely sensitive to intraperitoneal infection by VSV (Indiana serotype), which could be detected in lung tissues. This phenomenon has been still not illustrated yet. Secondly, according to our experimental results, the lethality of the recombinant virus expressing SARS-CoV-2 Delta S protein (rVSVΔG-Sdel-eGFP) was initially found to be decreased in Syrian hamsters, and further studies will be conducted to determine this phenomenon in the future.
Point 8: Lines 338-352 and 358-359, this assay holds the same weakness as the immunofluorescence (Figure 2B), the authors claim that this delta recombinant virus could be used as a surrogate virus of the authentic virus variant, even though they stated that “there was no significant difference between the detection results of neutralizing antibodies of two viruses and those of authentic SARS-CoV-2 and Delta variant” (line 343-344), they did not mention and not show the results with the authentic virus, they used a recombinant virus expressing Wuhan variant as a control virus. However, to warrant the neutralization assays are equivalent, they must run side-by-side their recombinant viruses with the authentic virus. Furthermore, the authors supported their claim that this recombinant virus is a surrogate for this assay because they detected lesser titers of antibodies in the samples tested (from humans, mice, and horses) for delta recombinant virus in comparison with Wuhan recombinant virus and this finding is accordingly with other reports, we do not exactly how these results could be using the authentic virus.
Response 8: We apologize for not providing a more detailed explanation in the section of "3.5 Establishment of a surrogate virus neutralization assay". According to your suggestion, we have supplemented the experiments related to the authentic virus, and results are showed in lines 359-365, 368-371 and Figure 5. The statistical analysis methods have been provided in the section of “Materials and Methods”.
Point 9: Line 379, the lack of lethality in the challenge assay in Syrian hamsters, which is a very common infection model described as SARS-CoV-2 infection, weakness the findings in this part of the paper, because one of the possibilities was to use this recombinant virus as a potential vaccine and full potentiality could not be tested, this should be further discussed.
Response 9: Thanks for your valuable suggestions. Syrian hamster is an ideal model for the SARS-CoV-2, which causes mild or moderate clinical symptoms. (PMID: 32408338, PMID: 32215622). Therefore, changes in body weight and viral load may be a better indicator of SARS-CoV-2 challenge. Interestingly, compared with rVSVΔG-Sdel-eGFP-vaccinated hamsters, even though high level of weight loss, viral load and RNA copies could be detected in PBS-vaccinated hamsters after challenged with the SARS-CoV-2 Delta variant, the titer of neutralizing antibody is increased in PBS-vaccinated hamsters. We have added the discussion of recombinant virus pathogenicity in the section of “Discussion”. (Lines 515-519)
[Minor comments]
Point 1: Line 47, develop the acronym ICU.
Response 1: We regret that there are problems with the acronym. The manuscript has been carefully revised and checked. “ICU” has been changed to “intensive care unit (ICU)”. (Line 47)
Point 2: Lines 57-58, develop the acronyms EBOV and MARV.
Response 2: “EBOV” and “MARV” have been changed to” Ebola virus (EBOV)” and “Marburg virus (MARV)”. (Lines 57-58)
Point 3: Line 124, develop the acronym EP.
Response 3: “EP tube” has been changed to “centrifuge tube”. (Line 127)
Point 4: Line 134, develop the acronym NC.
Response 4: “NC membrane” has been changed to “nitrocellulose filter membrane”. (Lines 138-139)
Point 5: Line 135, what does “closed” mean? I think “incubated” is a better option.
Response 5: According to your suggestion, we have changed “closed” to “incubated”. (Line 139)
Point 6: Line 135, describe the way that the polyclonal antibody was obtained.
Response 6: The information of antibody has been showed in the section of “Materials and Methods”. “Rabbit anti-SARS-CoV-2 S polyclonal primary antibody was purchased from Sino Bio-logical Inc.” (Lines 105-106)
Point 7: Line 137, given that the acronym HRP was developed at line 106, in this line “horseradish peroxidase ( )” needs to be deleted.
Response 7: “Horseradish peroxidase ( )” has been deleted. (Line 142)
Point 8: Line 140, the revealing substrate is missing (I assume is a reagent with luminol).
Response 8: According to your suggestion, we have added electrochemiluminescence reagent. (Lines 145-146)
Point 9
-Line 145, a capital letter is missing after “48h.”
Response 9
We have carefully revised similar problems throughout the text. We have corrected it. (Line 151)
Point 10: Line 165-166, I consider “calculated” a more appropriate word instead of “detected”.
Response 10: According to your suggestion, “detected” has been changed to “calculated”. (Line 172)
Point 11: Line 168, given that the neutralizing assays are performed to test the functional activity of antibodies, it is more suitable to say, “Neutralizing assay using the recombinant viruses”.
Response 11: Thanks for your valuable suggestion, “2.9. Neutralizing assay of recombinant viruses” has been changed to“2.9. Neutralizing assay using the recombinant viruses”. (Line 174)
Point 12: Line 193, I consider that “variant” is the proper word than “strain”.
Response 12: According to your suggestion, “strain” has been changed to “variant”. (Lines 199-200)
Reviewer 2 Report
The authors propose an experimental model consisting of vesicular stomatitis virus to study the viral host interactions of SARS CoV 2. Overall the manuscript is scientifically sound. However, there are some issues.
Abstract Ok
Introduction: A few phrases about virus quantification should be added
Materials and methods: a flow chart should be added.
Results: Some parts which belong to materials and methods are mixed with the results. The section should be reorganized. (line 226 to 259 should be moved to the materials and methods section. In the results section only results should be presented)
Discussions: A discussion of other experimental models used to study SARS-CoV-2 should be added.
Author Response
Response to Reviewer 2 Comments
Thank you for spending time in reviewing our manuscript and providing us with a list of constructive comments. Those comments are all valuable and very helpful for revising and improving our paper, as well as the important guiding significance to us. We have studied comments carefully and have made correction which we hope meet with approval.
Here are the responses to reviewer 2:
Point 1: Introduction: A few phrases about virus quantification should be added
Response 1: Thanks for your suggestion. The titer of rVSVΔG-Sdel-eGFP has been added in line 74. Meanwhile, "2.13. Quantification of Viral Loads by TCID50" has been explained in detail for better understanding. (Lines 223-228)
Point 2: Materials and methods: a flow chart should be added.
Response 2: A detailed flow chart of the full-length plasmid construction and the rescue of the recombinant virus has been presented in Figure 1. (Lines 124-130)
Point 3: Results: Some parts which belong to materials and methods are mixed with the results. The section should be reorganized. (lines 226 to 259 should be moved to the materials and methods section. In the results section only results should be presented)
Response 3: We have deleted the results belonging to materials and methods and moved them to the section of "2.3. Construction and rescue of recombinant viruses" for integration. (Lines 113-117, Lines 243-244)
Point 4: Discussions: A discussion of other experimental models used to study SARS-CoV-2 should be added.
Response 4: According to your suggestion, we have discussed other animal models of SARS-CoV-2 infection in the section of “Discussion” and finally chosen the Syrian hamster as the study subject, which is are economic, easy-operated and accessible and represents a mild or moderate model of SARS-CoV-2 infection. (Lines 484-519)
Reviewer 3 Report
- Major comments:
In this study, Wenwen He et al. characterized a replication-competent recombinant virus carrying the S protein of the SARS-CoV-2 Delta variant based on the vesicular stomatitis virus (VSV). The recombinant virus showed a replication advantage in Vero E6 cells with high titers. Besides, the authors also showed that the Delta variant spike proteins inside the VSV-vectored recombinant platform could mediate higher fusion activity and syncytium formation than the wild-type strain. In addition, the authors' data showed that the recombinant virus was avirulent in BALB/c mice, Syrian hamsters, 3-day ICR suckling mice, and IFNAR/GR-/- mice. It could also induce protective neutralizing antibodies in rodents and protect the Syrian hamsters against the SARS-CoV-2 Delta variant infection.
The recombinant SARS-CoV-2 Delta variant virus could help us understand the pathogenesis of the SARS-CoV-2 Delta variant, and the recombinant virus itself could be a valuable tool for coronavirus research.
The study is relevant to the field and well-organized.
- General concept comments
Here are some considerations for the study:
1. Will the S protein of the SARS-CoV-2 Delta variant accumulate mutations inside the animals since the recombinant virus is replication-competent? A sequence analysis should be provided to show the results, as the mutations, especially harmful ones, could be worrisome in the circumstance of the pandemic.
2. Since the delta variant is generally considered more virulent, and the authors also showed that rVSVΔG-Sdel-eGFP infection exhibited an S protein-mediated entry and induced cell-cell fusion much stronger than rVSVΔG-WH01-eGFP, the Growth kinetics of rVSVΔG-Sdel-eGFP should also be compared to rVSVΔG-WH01-eGFP.
The review could be further improved by including the following suggestions or considerations listed in specific comments.
- Specific comments:
1) Line 57, the full name of “EBOV” should be given.
2) For BALB/c mice, why only include the female ones?
3) Line 102 and 103, the accession number or strain name of the Wuhan-Hu-1 isolate, and Omicron strain S protein should be provided. Besides, no result was shown related to rVSVΔG-Omi-eGFP in the main result section.
4) Line 112, grammar problem, “The glycoprotein protein gene of p3.1-VSV-eGFP was replaced by the SARS-CoV-2 Delta variant spike protein gene…”
5) Lines 113 and 193, the accession number or strain name of the SARS-CoV-2 Delta variant spike should be provided.
6) Line 122, what types of cell lines were used for Virus Titration?
7) Line 124, how many diluted viruses were added to 96-well cell culture plates?
8) Line 137, no washing steps before the incubation of secondary antibodies?
9) Line 171, no data was shown for rVSVΔG-Omi-eGFP.
10) Line 211, the TCID50 was determined as 2.4. Virus Titration (Line 121)?
11) Line 236, would the rescued recombinant virus also have some rVSV-eGFP for the first round of infection since the pcDNA3.1-VSV-G helper plasmids were used?
12) Line 241, the viral titer was only 10 TCID50/mL in BSR/T7 cells?
13) Line 268, Figure 1C, the y axis was not in Log scale?
14) Line 268, Figure 1D, what is the scale bar of the image and what does “P” mean in “P1” and “P10”? Passage?
15) Line 284-285, suggests rephrasing as “the parental virus and the rVSVΔG-Sdel-eGFP morphological changes in cell culture supernatant were observed under transmission electron microscopy…”
16) Line 313, why the Pathogenicity of rVSVΔG-Sdel-eGFP in vivo was not compared to rVSVΔG-WH01-eGFP?
17) Line 345, no neutralizing antibodies results were shown for authentic SARS-CoV-2 and Delta variants in this section.
18) Line 346, although the authors claimed that the neutralization activity of sera or IgG against Delta strains was significantly reduced compared to the rVSVΔG-WH01-eGFP, there was no statistical analysis to support it.
19) Line 370, what surrogate virus neutralization assay was used?
20) Line 381 and Figure 5D, the statistical analysis result should be illustrated in Figure 5D.
21) Line 400 and Figures 6A and B, enlarged views are needed for a better illustration of the results.
22) Line 424-435, the grammar of this paragraph should be thoroughly checked.
23) Line 480, the new crown epidemic?
Author Response
Response to Reviewer 3 Comments
Thank you for spending time in reviewing our manuscript and providing us with a list of constructive comments. Those comments are all valuable and very helpful for revising and improving our paper, as well as the important guiding significance to us. We have studied comments carefully and have made correction which we hope meet with approval.
Here are the responses to reviewer 3:
[General concept comments]
Point 1: Will the S protein of the SARS-CoV-2 Delta variant accumulate mutations inside the animals since the recombinant virus is replication-competent? A sequence analysis should be provided to show the results, as the mutations, especially harmful ones, could be worrisome in the circumstance of the pandemic.
Response 1: Thanks for your constructive comments. The genome of recombinant virus of the tenth passage has been sequenced. There are no extra mutations or insertion/deletion mutations in the open reading frame of the S protein gene, and the result has been illustrated in lines 251-253. In addition, the replication-competent recombinant virus with multiple modifications to S, including truncation, mutation and et al is used to assess S entry requirement under BSL-2 conditions (PMID: 33353101, PMID: 32738193).
Point 2: Since the delta variant is generally considered more virulent, and the authors also showed that rVSVΔG-Sdel-eGFP infection exhibited an S protein-mediated entry and induced cell-cell fusion much stronger than rVSVΔG-WH01-eGFP, the Growth kinetics of rVSVΔG-Sdel-eGFP should also be compared to rVSVΔG-WH01-eGFP.
Response 2: Thanks for your insightful comment. We regret that the parallel comparison is not conducted due to the limitation of our study design. Since our preliminary experiments were designed to investigate the optimal production conditions of rVSVΔG-Sdel-eGFP and whether its growth kinetics changed after replacement of the exogenous protein gene, we did not explore the differences in growth kinetics between rVSVΔG-WH01-eGFP and rVSVΔG-Sdel-eGFP. The valuable suggestions you gave will be further explored in our future studies.
[Specific comments]
Point 1: Line 57, the full name of “EBOV” should be given.
Response 1: We regret that there were problems with the acronym. The manuscript has been carefully and thoroughly revised and checked. “EBOV” and “MARV” have been changed to “Ebola virus (EBOV)” and “Marburg virus (MARV)”. (Lines 57-58)
Point 2: For BALB/c mice, why only include the female ones?
Response 2: The female BALB/c mice is commonly used as animal model in laboratory. Firstly, we only selected female mice in the experimental design for consistency. Secondly, the female mice were friendlier and easier to handle. Thirdly, the sex was not considered in this study, and we will explore it in subsequent studies. In addition, most of studies about SARS-CoV-2 use female mice. (PMID:35746599, PMID: 32798445)
Point 3: Line 102 and 103, the accession number or strain name of the Wuhan-Hu-1 isolate, and Omicron strain S protein should be provided. Besides, no result was shown related to rVSVΔG-Omi-eGFP in the main result section.
Response 3: Thank you for your comments. We have provided information about the Wuhan-Hu-1 strain S protein gene and the accession number of the Wuhan-Hu-1 strain S protein gene is NC_045512.2. In addition, the accession number of the SARS-CoV-2 Delta variant S protein gene is OL336792.1. We apologize for adding content that is not relevant to the study, and we have removed "rVSVΔG-Omi-eGFP". (Lines 102-105)
Point 4: Line 112, grammar problem, “The glycoprotein protein gene of p3.1-VSV-eGFP was replaced by the SARS-CoV-2 Delta variant spike protein gene…”
Response 4: Thank you for the detailed review. We have carefully and thoroughly proofread the manuscript to correct all the grammar and typos. “The glycoprotein protein gene of p3.1-VSV-eGFP replaced by the SARS-CoV-2 Delta variant spike protein gene…” has been changed to “The glycoprotein protein gene of p3.1-VSV-eGFP was replaced by the SARS-CoV-2 Delta variant spike protein gene…”. (Lines 113-114)
Point 5: Lines 113 and 193, the accession number or strain name of the SARS-CoV-2 Delta variant spike should be provided.
Response 5: The accession number of the SARS-CoV-2 Delta variant S protein gene and the detail information of the SARS-CoV-2 Delta variant has been provided. The accession number of the SARS-CoV-2 Delta variant S protein gene is OL336792.1. The information of SARS-CoV-2 Delta variant (CSTR.16698.06. NPRC 6.CCPM-B-V-049-2105-8) has been presented in line 104. (Lines 104-105, Line 114)
Point 6: Line 122, what types of cell lines were used for Virus Titration?
Response 6: Thanks for your comment. The cell line has been provided in line 132. Vero E6 cells were used for Virus Titration.
Point 7: Line 124, how many diluted viruses were added to 96-well cell culture plates?
Response 7: The related information has been provided in line 134. The diluted viruses (100ul/well) were added to 96-well cell culture plates.
Point 8: Line 137, no washing steps before the incubation of secondary antibodies?
Response 8: Before the incubation of secondary antibodies, the membrane was washed five times with PBST. (Lines 148-149)
Point 9: Line 171, no data was shown for rVSVΔG-Omi-eGFP.
Response 9: We have removed "rVSVΔG-Omi-eGFP". We will check the full text and revise it.
Point 10: Line 211, the TCID50 was determined as 2.4. Virus Titration (Line 121)?
Response 10: To enable readers to better understand, we have added a more detailed description in the section of "2.13. Quantification of Viral Loads by TCID50". (Line 213-228)
Point 11: Line 236, would the rescued recombinant virus also have some rVSV-eGFP for the first round of infection since the pcDNA3.1-VSV-G helper plasmids were used?
Response 11: Due to the replacement of the VSV G protein by an exogenous protein gene in the full-length plasmid, a few numbers of rVSV-eGFP may be generated in the first generations, but these viruses will disappear after two cycle generations of culture.
Point 12: Line 241, the viral titer was only 10 TCID50/mL in BSR/T7 cells?
Response 12: Yes, we saw very little evidence for virus infection and/or virus production in BHK-21 cells (baby hamster kidney 21), and it probably because the cells express low levels of ACE2 mRNA (PMID: 33347434). Previous studies also supported our findings (PMID: 32990161, PMID: 32735849, PMID: 33347434).
Point 13: Line 268, Figure 1C, the y axis was not in Log scale?
Response 13: We have modified the y-axis in Figure 2B according to the comment. The y axis of Figure 2B has been changed to “rVSV titer (Log10 TCID50/mL)”. (Line 277)
Point 14: Line 268, Figure 1D, what is the scale bar of the image and what does “P” mean in “P1” and “P10”? Passage?
Response 14: First of all, we used the microscope magnification because we were unable to provide a scale bar due to the limitations of our lab equipment. Secondly, we apologize for not providing the meaning of "P" in the article. "P" represents "Passage", and we illustrate it in the figure legend. (Line 282)
Point 15: Line 284-285, suggests rephrasing as “the parental virus and the rVSVΔG-Sdel-eGFP morphological changes in cell culture supernatant were observed under transmission electron microscopy…”
Response 15: Thanks for your valuable comment. This sentence was rephrased according to the suggestion. We have changed it to “the parental VSV and the rVSVΔG-Sdel-eGFP morphological changes in cell culture supernatant were observed under transmission electron microscopy”. (Line 296-297)
Point 16: Line 313, why the Pathogenicity of rVSVΔG-Sdel-eGFP in vivo was not compared to rVSVΔG-WH01-eGFP?
Response 16: Thanks for your constructive comment. This problem is indeed a limitation of the study. The purpose of experiment is mainly to investigate the safety of rVSVΔG-Sdel-eGFP, so we only use rVSV-eGFP as a control. Further studies will be conducted to compare the pathogenicity of rVSVΔG-Sdel-eGFP and rVSVΔG-WH01-eGFP in vivo in the future.
Point 17: Line 345, no neutralizing antibodies results were shown for authentic SARS-CoV-2 and Delta variants in this section.
Response 17: According to your suggestion, we have supplemented the relevant experiments and the data have been presented in lines 359-363, lines 368-371 and Figure 5. Simultaneously, there are abundant studies to support our data (PMID:32738193, PMID:32990161, PMID:32735849).
Point 18: Line 346, although the authors claimed that the neutralization activity of sera or IgG against Delta strains was significantly reduced compared to the rVSVΔG-WH01-eGFP, there was no statistical analysis to support it.
Response 18: Thanks for your insightful comment. We are sorry for the lack of detailed description in the section of "3.5 Establishment of a surrogate virus neutralization assay", and we have added the statistical results and p value in line 363. Meanwhile, a new statistical method is incorporated into the section of " 2.15. Statistical analysis". (Lines 238-239)
Point 19: Line 370, what surrogate virus neutralization assay was used?
Response 19: “Surrogate virus” is the VSV-based replication-competent recombinant viruse carrying an eGFP tag (rVSVΔG-Sdel-eGFP). “Surrogate virus neutralization assay” is the assay using the recombinant virus to test neutralizing antibody activities.
Point 20: Line 381 and Figure 5D, the statistical analysis result should be illustrated in Figure 5D.
Response 20: The results of statistical analysis are presented in detail in Figure 6D, and the statistical method is Two-way ANOVA analysis. (Line 398)
Point 21: Line 400 and Figures 6A and B, enlarged views are needed for a better illustration of the results.
Response 21: Thank you for your suggestion. We are very sorry for our negligence of the clarity of the pictures, and we have enlarged Figures 7A and B. Characteristic pathological changes with arrows are pointed out in the figures.
Point 22: Line 424-435, the grammar of this paragraph should be thoroughly checked.
Response 22: We regret there are problems with the English. The manuscript has been carefully revised and checked by a native English speaker to improve the grammar and readability. (Line 444-454)
Point 23: Line 480, the new crown epidemic?
Response 23: We are very sorry for this grammatical error and will check the grammatical issues in the full text more carefully. This description has been revised as “the SARS-CoV-2 pandemic”. (Line 529)
Round 2
Reviewer 1 Report
Even though the experimental design for Figure 5 was not suitable to demonstrate that is a surrogate neutralization assay since the authentic virus was not run in parallel with the recombinant viruses, I acknowledge the authors to let me know this during their responses. I consider that this should be added in the material and methods section, saying something like this:
2.9 Neutralizing assay using the recombinant viruses
....-eGFP were tested, in previous experiments, the authentic virus (SARS-CoV-2 Delta variant) was used in the same conditions of the experiment.
In the present form seems that all viruses were tested in parallel, even a correlation analysis was added. I suggest adding all the references as support that this system has been used in neutralization assays with successful results (these were given by authors in the response letter: PMID 32738193, 32990161, 32735849).
Author Response
Thank you for spending time in reviewing our manuscript and providing us some constructive comments. We have made correction which we hope meet with approval.
Here is the response to reviewer 1:
Point 1: Even though the experimental design for Figure 5 was not suitable to demonstrate that is a surrogate neutralization assay since the authentic virus was not run in parallel with the recombinant viruses, I acknowledge the authors to let me know this during their responses. I consider that this should be added in the material and methods section, saying something like this:
2.9 Neutralizing assay using the recombinant viruses
....-eGFP were tested, in previous experiments, the authentic virus (SARS-CoV-2 Delta variant) was used in the same conditions of the experiment.
In the present form seems that all viruses were tested in parallel, even a correlation analysis was added. I suggest adding all the references as support that this system has been used in neutralization assays with successful results (these were given by authors in the response letter: PMID 32738193, 32990161, 32735849).
Response 1: Thanks for your valuable suggestions. According to your advice, we have added “in previous experiments, the authentic virus (SARS-CoV-2 Delta variant) was used in the same conditions of the experiment” in the section of “2.9 Neutralizing assay using the recombinant viruses” (lines 184-186). Meanwhile, the references (PMID 32738193, 32990161, 32735849) have been cited in lines 372-374. The amendments are highlighted in yellow in the revised manuscript.

Reviewer 2 Report
The manuscript has been improved significantly. The authors have answered all the comments.
Author Response
Thanks for the positive comments.
Reviewer 3 Report
I think that the manuscript has been improved, and the authors have addressed most of my concerns.

Author Response
Thanks for the positive comments